# Vitamin C, Hydrocortisone, and Thiamine for the Treatment of Severe Sepsis and Septic Shock: A Retrospective Analysis of Real-World Application

**DOI:** 10.3390/jcm8040478

**Published:** 2019-04-09

**Authors:** Jane J. Litwak, Nam Cho, H. Bryant Nguyen, Kayvan Moussavi, Thomas Bushell

**Affiliations:** 1Department of Pharmacy, HonorHealth John C. Lincoln Medical Center, 250 E. Dunlap Ave., Phoenix, AZ 85020, USA; jane.litwak@gmail.com; 2Department of Pharmacy Practice, Loma Linda University School of Pharmacy, 24745 Stewart Street, Loma Linda, CA 92350, USA; nacho@llu.edu (N.C.); hbnguyen@llu.edu (H.B.N.); 3Department of Pharmacy, Loma Linda University Medical Center, 11234 Anderson Street, Loma Linda, CA 92354, USA; 4Division of Pulmonary, Critical Care, Hyperbaric, Allergy and Sleep Medicine, Department of Medicine, Loma Linda University School of Medicine, 11234 Anderson St, Loma Linda, CA 92354, USA; 5Department of Pharmacy Practice, Marshall B. Ketchum University College of Pharmacy, 2575 Yorba Linda Blvd., Fullerton, CA 92831, USA; kmoussavi@ketchum.edu

**Keywords:** sepsis, septic shock, vitamin C, hydrocortisone, thiamine

## Abstract

A recent study suggested mortality benefits using vitamin C, hydrocortisone, and thiamine combination therapy (triple therapy) in addition to standard care in patients with severe sepsis and septic shock. In order to further evaluate the effects of triple therapy in real-world clinical practice, we conducted a retrospective observational cohort study at an academic tertiary care hospital. A total of 94 patients (47 in triple therapy group and 47 in standard care group) were included in the analysis. Baseline characteristics in both groups were well-matched. No significant difference in the primary outcome, hospital mortality, was seen between triple therapy and standard care groups (40.4% vs. 40.4%; *p* = 1.000). In addition, there were no significant differences in secondary outcomes, including intensive care unit (ICU) mortality, requirement for renal replacement therapy for acute kidney injury, ICU length of stay, hospital length of stay, and time to vasopressor independence. When compared to standard care, triple therapy did not improve hospital or ICU mortality in patients with septic shock. A randomized controlled trial evaluating the effects of triple therapy is necessary prior to implementing vitamin C, hydrocortisone, and thiamine combination therapy as a standard of care in patients with septic shock.

## 1. Introduction

Sepsis is a serious disease state affecting 15–19 million people globally every year [1]. In the United States alone, the annual incidence of sepsis exceeds 1.5 million cases, with mortality rates reaching up to 30% and 45% for sepsis and septic shock, respectively [2,3]. In low and middle-income countries, the incidence of sepsis is estimated to be even higher due to widespread malnutrition, higher rates of infection, and lack of critical care resources [4,5]. There is urgent need for cost-effective and easily accessible therapies for sepsis and septic shock. A recent retrospective before-and-after study exploring the use of triple therapy (TT) with 1.5 g intravenous (IV) vitamin C every six hours (for four days or until intensive care unit (ICU) discharge), 50 mg IV hydrocortisone every six hours (for seven days or until ICU discharge), and 200 mg IV thiamine every 12 h (for four days or until ICU discharge) in severe sepsis and septic shock patients demonstrated mortality benefits [6]. In this study, Marik et al. found that patients who received TT had 8.5% hospital mortality compared to 40.4% hospital mortality in the control arm (*p* < 0.001) [6]. Moreover, patients receiving TT had significantly shorter time to vasopressor independence, lower requirement for renal replacement therapy (RRT), and a greater decrease in Sepsis-Related Organ Failure Assessment (SOFA) score and procalcitonin (PCT) level. The authors concluded that the early use of IV vitamin C, corticosteroids, and thiamine is effective in preventing progressive organ dysfunction, including acute kidney injury (AKI), and in reducing mortality in patients with severe sepsis and septic shock [6]. A recent review article by Moskowitz et al. further discusses the physiological rationale and existing data supporting the use of TT in sepsis [7]. 

While these preliminary findings are promising and merit further research, clinicians at our institution and others have begun to adopt the use of TT as an additional treatment option available for septic shock. In this study, we performed a retrospective analysis of our real-world experience to further evaluate the effects of TT in patients with septic shock.

## 2. Materials and Methods

### 2.1. Patient Identification and Intervention

This was an Institutional Review Board (IRB)-approved, retrospective cohort study conducted at a tertiary, academic, level 1 trauma center with 507 beds. Consecutive adult patients with International Classification of Disease (ICD)-10 code for “septic shock” admitted to medical, surgical, or neurocritical care ICUs between October 28, 2016 and June 10, 2018 were identified via electronic health records (EHR—Epic Systems Corporation, Verona, WI, USA). Among the patients identified, only the patients receiving vasopressor therapy were consecutively included in the treatment group if the patients received TT, and in the control group if patients did not receive TT. Pregnant patients were excluded. 

At our institution, treatment of sepsis, severe sepsis, and septic shock is in accordance with the Center for Medicare and Medicaid Services (CMS) and Joint Commission Early Management Bundle, Severe Sepsis/Septic Shock (SEP-1), which is consistent with the treatment bundle recommended by the 2012 Surviving Sepsis Campaign (SSC) guidelines [8,9]. Per SEP-1 and the 2012 SSC guideline bundle, initial fluid resuscitation begins with 30 mL/kg (or 2 liters) of crystalloid and administration of broad spectrum antibiotics within 3 h of diagnosis of severe sepsis and septic shock. Vasopressor therapy is initiated if hypotension persists despite adequate fluid resuscitation. Norepinephrine is the vasopressor of choice and is typically titrated to a maximum rate of 50 mcg/min to maintain a mean arterial pressure (MAP) > 65 mmHg. All patients included in the study were treated with standard care consisting of fluids, vasopressors, and empiric broad-spectrum antibiotics. Patients were included in the TT group if they received at least one dose of each of the following medications intravenously: 1.5 g vitamin C every 6 h, 200–300 mg hydrocortisone daily (50 mg every 6 h or 100 mg every 8 h), and thiamine 200 mg every 12 h. The standard care (SC) group consisted of patients receiving standard therapy alone; however, patients in this standard care group could receive hydrocortisone. The first 47 patients that met the inclusion criteria for each group were included for statistical analysis.

### 2.2. Outcomes and Data Analysis

A review of each patient’s EHR provided clinical and demographic data, including age, sex, admitting diagnosis, comorbidities, requirement for mechanical ventilation, duration of vasopressors, Acute Physiology and Chronic Health Evaluation (APACHE) II score, APACHE IV score, ICU and hospital length of stay (LOS), ICU and hospital survival, laboratory data, and medication use [10,11]. The SOFA score, serum creatinine, white blood cell (WBC) count, platelet count, total bilirubin, PCT, and lactate levels were recorded from the first day of vasopressor therapy and daily thereafter for a total of 4 days [12].

Patients were considered immunocompromised if they received cytotoxic therapy within 30 days of admission or were receiving immunosuppressant therapy for organ transplantation. Acute kidney injury (AKI) was defined as an increase in serum creatinine ≥ 0.3 mg/dL within 48 h or a level ≥ 1.5 times the baseline value per Kidney Disease: Improving Global Outcomes (KDIGO) criteria [13]. Patients with unknown baseline serum creatinine with initial admission serum creatinine > 1.5 mg/dL were regarded as having AKI. Appropriate antimicrobial therapy was defined as the selection of antimicrobials that have in vitro activity against the primary isolated pathogens as denoted in 2018 Sanford Guide to Antimicrobial Therapy [14]. However, if the patient’s microbiological data revealed organisms’ resistance to the selected antimicrobials, these antimicrobials were deemed inappropriate. In patients that were culture negative, antimicrobials were deemed appropriate if they had in vitro activity against common causative organisms for a given source of infection, as denoted in 2018 Sanford Guide to Antimicrobial Therapy [14]. Timeliness of antimicrobial use and fluid therapy was defined as the administration of broad-spectrum antibiotics and fluid boluses 3 h before and after the time of sepsis diagnosis. For the purposes of this study, we used the time at which the first culture was ordered as a surrogate for the time of diagnosis of sepsis. 

The primary outcome was hospital mortality. Secondary outcomes were ICU mortality, ICU and hospital LOS, duration of vasopressor therapy, requirement for RRT in patients with AKI, and changes in serum creatinine, PCT, lactate, and SOFA scores within the first 72 h of initiation of vasopressor therapy. Vasopressor independence was defined as freedom from vasopressor use for more than 48 h. 

All categorical data are presented as numbers and percentages and were compared using chi-square or Fisher’s exact test, when appropriate. Parametric continuous data are presented using means and standard deviations while non-parametric continuous data are presented using median and interquartile range. Continuous data were compared using Student’s *t*-test or Mann-Whitney U test, if parametric or non-parametric, respectively. Fifty-four patients (27 patients in each group) will provide 80% power to detect a 31.9% difference in mortality as seen in the Marik et al. study with a significance level of 0.05 [6]. All statistical analyses were performed with SPSS Statistics version 23 (IBM, Chicago, IL, USA).

## 3. Results

While our power analysis required 54 patients, we enrolled 94 patients (47 patients in each group) in an attempt to replicate the methods and results by Marik et al. [6]. However, twenty-seven patients (57.4%) did not complete TT treatment for a full four days or until ICU discharge due to following reasons: therapy discontinuation by primary team (*n* = 9 (33.3%)), patient death (*n* = 9 (33.3%)), missed doses due to medication delivery issues, line access issues, operations (*n* = 5 (18.5%)), insufficient number of doses ordered (*n* = 3 (11.1%)), and thiamine shortage (*n* = 1 (3.7%)). Baseline characteristics were well-matched between the two groups (Table 1). The most common infection in both groups was pneumonia (TT 46.8% vs. SC 38.3%; *p* = 0.404), followed by gastrointestinal and biliary infection (TT 21.3% vs. SC 31.9%; *p* = 0.243). Sixteen patients (34%) in the treatment group and 18 patients (38.3%) in the SC group had positive blood cultures (*p* = 0.668). 

Both groups were similarly treated with fluids and antibiotics per sepsis treatment guidelines. Thirty-one patients (66%) in the TT group received fluids within three hours of diagnosis of sepsis compared with 36 patients (76.6%) in the SC group (*p* = 0.254) (Table 1). Of those patients, 21 patients (67.7%) in the TT group and 23 patients (63.9%) in the SC group received at least 30 mL/kg of fluids (*p* = 0.740). Thirty-two patients (68.1%) in the TT group and 26 patients (55.3%) in the SC group received antibiotics (*p* = 0.203). Both groups had similar rates of receiving appropriate antibiotics within three hours of diagnosis of sepsis (TT 78.7% vs. SC 70.2%; *p* = 0.344). In the TT group, the mean number of doses of vitamin C and thiamine were 12.3 ± 6.3 and 6.6 ± 3.2 per patient, respectively. The mean daily dose of hydrocortisone was 176.7 ± 42.0 mg, and the median duration of therapy was 96 h (48–156 h). The median time to TT initiation was 13.8 h (7.5–36.8 h) from vasopressor initiation. Nineteen patients (40.4%) in the SC group were treated with hydrocortisone. The mean daily dose of hydrocortisone was 177.5 ± 42.5 mg, and the median duration of therapy was 104.5 h (77.3–188.0 h). Compared to the TT group, the mean daily dose of hydrocortisone was similar in the SC group (*p* = 0.941).

There was no difference in hospital mortality between groups (TT 40.4% vs. SC 40.4%; *p* = 1.000) (Figure 1, Table 2). Thirty-eight patients (80.9%) in the TT group and 32 patients (68.1%) in the SC group developed AKI (*p* = 0.156). Eleven patients (28.9%) with AKI in the TT group required RRT compared to 11 patients (34.4%) in the SC group (*p* = 0.626). There was no difference in time to vasopressor independence between the TT and SC groups (TT 84.2 h (37–169.3 h) vs. SC 62.5 h (32.6–105.9 h); *p* = 0.324). The SOFA score increased by 1.3 ± 4.1 in the TT group and by 0.1 ± 4.7 in the SC group within 72 h of vasopressor initiation (*p* = 0.390). The median 72-h change in PCT was 0.1 (−55.0–9.1) in the TT group and 2.5 (−3.2–4.4) in the SC group (*p* = 0.268). Median ICU LOS (TT 11 days (7–19 days) vs. SC 10 days (5–17 days); *p* = 0.491) and hospital LOS (TT 19 days (9–26 days) vs. SC 14 days (8–23 days); *p* = 0.346) were similar between groups (Table 2).

Twenty patients (43%) received TT for a full 4 days or until ICU discharge. Due to a large number of patients not completing the full treatment, a confirmatory analysis was performed in subgroup of patients who did. There was no difference in hospital mortality (TT 35% vs. SC 40.4%; *p* = 0.677), ICU mortality (TT 30% vs. SC 38.3%; *p* = 0.517), need for RRT for AKI (TT 31.3% vs. SC 34.4%; *p* = 0.829), ICU LOS (TT 14.5 days (7.5–22.0 days) vs. SC 10.0 days (5.0–17.0 days); *p* = 0.204), hospital LOS (TT 19.0 days (9.3–23.5 days) vs. SC 14.0 days (8.0–23.0 days); *p* = 0.376), or time to vasopressor independence (TT 85.1 h (42.8–176.2 h) vs. SC 62.5 h (32.6–105.9 h); *p* = 0.250) (Table 3). 

## 4. Discussion

In this retrospective study of real-world experience in patients with septic shock, the administration of TT (IV vitamin C, hydrocortisone, and thiamine) in addition to standard sepsis therapy did not improve mortality compared to standard therapy alone. TT did not decrease the need for RRT initiation in patients with AKI, nor did it shorten ICU LOS, hospital LOS, or the duration of vasopressor therapy when compared to SC. These findings align with the recently published results from Shin et al’s retrospective study of early TT treatment of patients with septic shock [15]. In this propensity score-based analysis of a before-and-after cohort study, Shin et al found that there was no mortality benefit in early administration of vitamin C and thiamine in the overall patient population; however, subgroup analysis revealed increased survival in patients with hypoalbuminemia (albumin < 3.0) or severe organ failure (SOFA >10) [15]. 

Our findings also differ from the results of Marik et al. which demonstrated significant decreases in mortality and end organ failure in patients with severe sepsis and septic shock who received TT [6]. There are several explanations that may rationalize the differences in outcomes. The TT regimen was not protocolized at our institution, and 27 patients (57%) in the TT group did not receive TT for the full treatment duration (four days or until ICU discharge as outlined in the Marik et al protocol). Of these patients, the most common reasons for insufficient duration of therapy were therapy discontinuation by the primary team or patient death. This may have diminished the overall benefit of TT in the treatment group. However, the confirmatory analysis of subgroup of patients that received full treatment duration revealed that none of the primary or secondary outcomes had significant differences versus SC.

In addition, because TT was not protocolized at our institution, the decision to initiate and the timing of discontinuation of TT was solely based upon provider preference. While some providers have adopted the use of TT, others have not. This could have been a confounding factor that led to our inability to find a difference between the two groups, as the overall management of septic shock may also be different between groups (i.e., clinicians who have adopted the use of TT may also start vasopressors earlier or later than clinicians who have not).

As a retrospective study, our analysis was limited by a large number of missing PCT values and SOFA scores. PCT and SOFA scores increased from day 1 to day 4 in both groups despite receiving sepsis treatment. This can be explained in part by insufficient lab values that may have inaccurately represented the overall group’s PCT and SOFA score changes. Inappropriate empiric antibiotic selection was observed in nearly 30% of patients overall, which could have also contributed to the increase in PCT. 

Finally, maximal benefit of TT is likely seen within 24 h of sepsis diagnosis and ICU admission as noted in the Marik et al. study [6]. TT may have been used as a last-line therapy in patients that were not clinically improving, which could explain why TT was found to have limited efficacy in our study. However, most patients (72%) received TT within 24 h of vasopressor initiation, indicating a majority of patients had TT initiated a short time after septic shock diagnosis. 

Although we were unable to reproduce the results found by Marik et al. [6] in our trial, a theoretical benefit for each of the medications included in TT does exist. Based on the mechanism of action of each of the individual medications included, their use can potentially be used in the treatment of patients with septic shock. Vitamin C (ascorbic acid) functions as a cofactor for biosynthesis of neurotransmitters such as noradrenaline, cortisol, and vasopressin, and produces anti-inflammatory effects in the body [16]. Plasma vitamin C levels are significantly depleted in critically ill patients, which can contribute to hypotension, exaggerated inflammation, microcirculatory compromise, oxidative organ injury, and impaired immune defense [16,17,18]. Treatment of hypovitaminosis has been shown to decrease SOFA scores, PCT, C-reactive protein, and thrombomodulin levels, which can translate to decreases in organ failure, inflammation, and endothelial injury [19]. Intravenous vitamin C doses of 2 g/day can normalize vitamin C plasma concentrations [20]. However, doses greater than 10 g/day should be avoided due to its association with supraphysiological plasma concentrations that lead to increased oxalate excretion and metabolic alkalosis [20].

Thiamine functions as a coenzyme for various biopathways such as glucose metabolism, Krebs cycle, and adenosine triphosphate (ATP) synthesis [16]. Thiamine deficiency is also common in critically ill patients and can lead to lactic acidosis due to pyruvate’s inability to enter the Krebs cycle [16]. Supplementation of 200 mg IV thiamine every 12 h has been shown to significantly decrease lactate levels in critically ill patients with baseline thiamine deficiency [21]. 

Hydrocortisone was first incorporated into the Surviving Sepsis Campaign guidelines in 2004 and continues to be recommended in the 2016 Update [22,23]. Evidence for hydrocortisone in septic shock is conflicting; however, there are theoretical benefits. Glucocorticoids can prevent the induction of nitric oxide synthase, an enzyme that can cause relaxation of the vascular smooth muscle and result in vasodilation and hypotension [24]. Glucocorticoids can also potentiate the effects of catecholamines via glucocorticoid receptors on vascular smooth muscle cells [25]. In vascular endothelial cells, glucocorticoids suppress the production of vasodilators such as prostacyclin and nitric oxide [25]. As an anti-inflammatory agent, hydrocortisone can also help mitigate the inflammatory responses and relative adrenal insufficiency during septic shock [26,27]. Along with vitamin C and thiamine, these three medications can theoretically work synergistically to reduce endothelial barrier changes caused by septic shock and prevent nephropathy caused by overproduction of oxalate [28,29]. Despite the conflicting evidence of hydrocortisone, Annane et al. demonstrated its mortality benefits in septic shock patients in two different randomized controlled trials [30,31]. In our study, fewer patients in the control group received hydrocortisone (40.4%) compared to those of Marik et al’s study (59.6%), yet mortality rates were equal between SC and TT groups [6]. This further strengthens the results of our study as there was less probability of hydrocortisone confounding the mortality benefits of the vitamin C, hydrocortisone, and thiamine combination therapy. 

There are several randomized controlled trials currently underway, exploring the use of vitamin C, hydrocortisone, and thiamine in various sepsis populations. Similar to our study, the Evaluation of Hydrocortisone, Vitamin C, and Thiamine for the Treatment of Septic Shock (HYVITS – NCT03380507) trial will be evaluating hospital mortality as a primary outcome in patients with septic shock. The HYVITS trial will also assess the safety of TT by evaluating the rates of nephrolithiasis, hemolysis, GI bleeding, blood glucose, hypernatremia, and hypokalemia.

## 5. Conclusions

In our study of patients with septic shock, administration of intravenous vitamin C, hydrocortisone, and thiamine combination therapy did not improve hospital mortality. Moreover, TT did not prevent the need for RRT, nor did it lower the duration of vasopressor therapy or reduce the length of ICU or hospital stay. These results suggest that incorporating vitamin C, hydrocortisone, and thiamine into standard practice may not improve patient outcomes, but rather may increase cost. Randomized controlled trials are necessary for further evaluation of the potential benefits of this therapy. Questions such as time and duration of therapy, indicated patient subgroups, and efficacy are to be answered before TT is considered standard care in the management of patients with severe sepsis and septic shock.

## Figures and Tables

**Figure 1 jcm-08-00478-f001:**
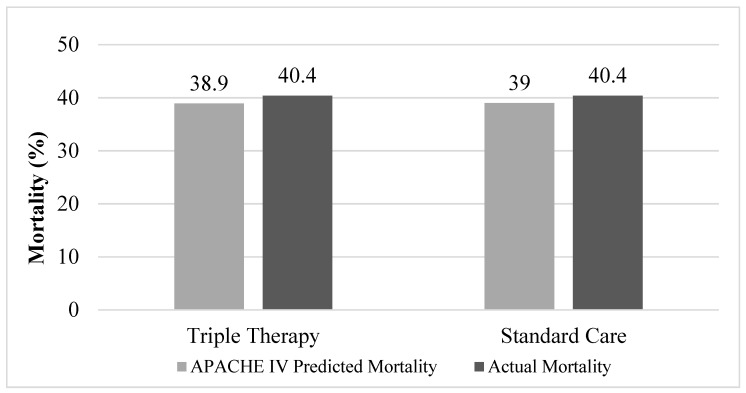
APACHE (Acute Physiology and Chronic Health Evaluation) intravenous (IV) predicted mortality compared to actual mortality. Actual mortality in triple therapy and standard care groups were the same at 40.4% (*p* = 1.000).

**Table 1 jcm-08-00478-t001:** Baseline characteristics.

Variables	Triple Therapy (*n* = 47)	Standard Care (*n* = 47)	*p*
Age, mean ± SD, year	58.3 ± 17.0	60.1 ± 14.0	0.589
Weight, mean ± SD, kg	82.1 ± 32.6	80.7 ± 22.5	0.812
Sex, male, No (%)	28 (59.6)	29 (61.7)	0.833
Comorbidities, No (%)			
None/Unknown	4 (8.5)	3 (6.4)	1.000
Diabetes	14 (29.8)	10 (21.3)	0.344
Hypertension	18 (38.3)	21 (44.7)	0.53
CAD/MI	7 (14.9)	4 (8.5)	0.336
Heart failure	5 (10.6)	4 (8.5)	0.503
Malignancy	8 (17.0)	12 (25.5)	0.313
COPD	2 (4.3)	7 (14.9)	0.158
Cirrhosis	5 (10.6)	11 (23.4)	0.100
CVA	5 (10.6)	1 (2.1)	0.203
CKD	4 (8.5)	10 (21.3)	0.082
Immunocompromised	3 (6.4)	6 (12.8)	0.486
Drug addiction	4 (8.5)	5 (10.6)	1.000
Primary diagnosis No (%)			
Pneumonia	22 (46.8)	18 (38.3)	0.404
Urosepsis	4 (8.5)	8 (17.0)	0.216
Primary bacteremia	3 (6.4)	3 (6.4)	1.000
GI/biliary	10 (21.3)	15 (31.9)	0.243
Other (meningitis, TSS, unknown, patient deceased before cultures, necrotizing fasciitis)	7 (14.9)	4 (8.5)	0.336
Unknown	1 (2.1)	0 (0.0)	1.000
Mechanical ventilation, No (%)	43 (91.5)	39 (83.0)	0.216
Vasopressors, No (%)	47 (100)	47 (100)	1.000
Acute kidney injury, No (%)	38 (80.9)	32 (68.1)	0.156
Positive blood cultures, No (%)	16 (34.0)	18 (38.3)	0.668
Lab values			
WBC, mean ± SD, ×10^9^ (excluding immunosuppressed patients)	16.6 ± 13.0	16.1 ± 11.8	0.834
Lactate, median (IQR), mmol/L	2.7 (1.5–5.5)	2.9 (1.5–4.2)	0.708
Creatinine, median (IQR), mg/dL (excluding CKD)	1.4 (0.9–2.2)	1.4 (0.9–2.5)	0.988
Procalcitonin, median (IQR), mcg/mL	7.3 (0.7–52.1)	4.3 (1.4–13.3)	0.534
Treatment timing and duration			
Fluids within 3 h of culture, No (%)	31 (66.0)	36 (76.6)	0.254
Fluids at least 30 mL/kg (within 3 h), No (%)	21 (67.7)	23 (63.9)	0.740
Antibiotics within 3 h of culture, No (%)	32 (68.1)	26 (55.3)	0.203
Appropriate antibiotics (overall), No (%)	37 (78.7)	33 (70.2)	0.344
Number of vitamin C doses, mean ± SD	12.3 ± 6.3	0	–
Duration of vitamin C therapy, median (IQR), h	90.0 (45.0–96.0)	0	–
Number of thiamine doses, mean ± SD	6.6 ± 3.2	0	–
Duration of thiamine therapy, median (IQR), h	96.0 (48.0–96.0)	0	–
Receipt of hydrocortisone, No (%)	47 (100)	19 (40.4)	<0.05
Daily dose of hydrocortisone, mean ± SD, mg	176.7 ± 42.0	177.5 ± 42.5	0.941
Duration of hydrocortisone therapy, median (IQR), h	96.0 (48.0–156.0)	104.5 (77.3–188.0)	0.220
Critical illness scores and predicted mortality			
Day 1 SOFA, mean ± SD	10.6 ± 10.6	9.7 ± 10.0	0.211
APACHE II, mean ± SD	21.5 ± 8.0	20.0 ± 7.4	0.739
APACHE IV, mean ± SD	88.6 ± 29.1	84.1 ± 25.4	0.455
APACHE IV Predicted mortality, mean ± SD	38.9 ± 27.2	39.0 ± 23.6	0.991

SD: standard deviation; CAD: coronary artery disease; MI: myocardial infarction; COPD: chronic obstructive pulmonary disease; CVA: cerebrovascular accident; CKD: chronic kidney disease; GI: gastrointestinal; TSS: toxic shock syndrome; WBC: white blood cell; IQR: interquartile range; SOFA: Sepsis-Related Organ Failure Assessment; APACHE: Acute Physiologic and Chronic Health Evaluation.

**Table 2 jcm-08-00478-t002:** Outcomes in patients who received triple therapy versus no triple therapy.

Variables	Triple Therapy (*n* = 47)	Standard Care (*n* = 47)	*p*
Hospital mortality, No. (%)	19 (40.4)	19 (40.4)	1.000
ICU mortality, No (%)	17 (36.2)	18 (38.3)	0.831
RRT for AKI, No. (%)	11 of 38 (28.9)	11 of 32 (34.4)	0.626
ICU LOS, median (IQR), days	11.0 (7.0–19.0)	10.0 (5.0–17.0)	0.491
Hospital LOS, median (IQR), days	19.0 (9.0–26.0)	14.0 (8.0–23.0)	0.346
Duration of vasopressors, median (IQR), h	84.2 (37.0–169.3)	62.5 (32.6–105.9)	0.324
ΔSOFA score in 72 h, mean ± SD	1.3 ± 4.1	0.1 ± 4.7	0.390
ΔPCT in 72 h, median (IQR), ng/mL	0.1 (−55.0–9.1)	2.5 (−3.2–4.4)	0.268

ICU: intensive care unit; AKI: acute kidney injury; RRT: renal replacement therapy; LOS: length of stay; IQR: interquartile range; SOFA: Sepsis-Related Organ Failure Assessment; PCT: procalcitonin.

**Table 3 jcm-08-00478-t003:** Subgroup analysis of patients who completed triple therapy (full four days or transferred out of the intensive care unit (ICU)) versus no triple therapy.

Variables	Triple Therapy (*n* = 20)	Standard Care (*n* = 47)	*p*
Hospital mortality, No. (%)	7 (35.0)	19 (40.4)	0.677
ICU mortality, No (%)	6 (30.0)	18 (38.3)	0.517
RRT for AKI, No. (%)	5 of 16 (31.3)	11 of 32 (34.4)	0.829
ICU LOS, median (IQR), days	14.5 (7.5–22.0)	10.0 (5.0–17.0)	0.204
Hospital LOS, median (IQR), days	19.0 (9.3–23.5)	14.0 (8.0–23.0)	0.376
Duration of vasopressors, median (IQR), h	85.1 (42.8–176.2)	62.5 (32.6–105.9)	0.250

ICU: intensive care unit; RRT: renal replacement therapy; AKI: acute kidney injury; LOS: length of stay; IQR: interquartile range.

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
