# Peer review of "Vitamin C, Hydrocortisone, and Thiamine for the Treatment of Severe Sepsis and Septic Shock: A Retrospective Analysis of Real-World Application"

_jcm, 2019, doi:10.3390/jcm8040478_

Round 1
Reviewer 1 Report
This study is a retrospective analysis of 94 total patients with septic shock receiving “Triple Therapy (TT)” (47 patients) or standard of care alone (47 patients) with the purpose of evaluating the effects of TT on sepsis-related outcomes. The determination of septic shock was gathered from discharge diagnosis data, and the standard of care is in alignment with various governing bodies. This current study found no difference in outcomes of those who received TT vs those who did not. Many of the patients included in this study did not complete the TT protocol as described by Marik et al for various reasons which were described, but not well quantified. There authors postulate that this may have played a role in the outcome of the study; however, an analysis of those who did complete the protocol also resulted in similar outcomes as the control group.
Overall I feel that this study, though negative, is both important and timely. Due to the data published by Marik et al, many clinicians, such as the authors, have, to some degree, adopted TT, even though many have expressed skepticism regarding the results. This study suggests that TT may have no beneficial impact; however, importantly, no detriment was detected. The study has strengths in that the TT and control groups were both similar, and the study certainly reflects a real-world environment. It is unclear, though, what drove the decision to use TT vs not. Do certain clinicians do so while others do not? If so, though difficult, controlling for the clinician may be necessary, as they may differ in other decisions as well (such as when to use vasopressors). If it is not only clinician preference on when to use TT, an explanation of this may be warranted. Also, the authors note that ‘consecutive’ patients were enrolled. It seems unusual that of the 94 patients included, exactly half received TT while others did not. A study flow diagram would provide some degree of perspective on how many patients were admitted to the ICU during the study period, and why they were excluded from analysis (for example, they did not go on to develop shock). It would also be helpful to reference the time of initiation of TT from the triage time to help indicate, as noted by the authors, in what percent of patients TT was a late addition to therapy.
Finally, I feel the tables to be generally helpful and easily-read.
Author Response
Dear Reviewer,
Please see attached document for responses to your comments and suggestions. Thank you for your time in reviewing and helping to improve our manuscript.
Thank you,
Thomas Bushell

Reviewer 2 Report
Summary
In this work the authors present the findings of a retrospective cohort study evaluating the efficacy of vitamin C, thiamine, and hydrocortisone “triple therapy” (TT) in addition to standard for patients with septic shock managed under real-world conditions. A total of 94 patients with septic shock based on ICD-10 coding data were consecutively included in the study (47 TT + standard care vs. 47 standard care alone) and outcomes such as hospital mortality, ICU mortality, utilization of dialysis for acute kidney injury, duration of vasopressors, change in SOFA score, and length of stay metrics were compared. The authors found no difference in any of the analyzed outcomes, in contrast to prior work published by Marik and colleagues in 2017, which demonstrated a dramatic reduction in mortality in patients with septic shock treated with TT + standard of care as compared to standard of care only. The authors acknowledge that the majority of patients in the experimental group did not complete the full TT protocol as described by Marik et al. and include a subgroup analysis of patients who did complete the full protocol, again demonstrating no differences in outcomes as compared to controls.
While this work is not novel, it provides important new data on the efficacy of this recently developed treatment regimen, which has been a point of some contention and controversy within the critical care community since it was first described. The study is methodologically well-constructed and benefits from a lack of major differences in patient demographics, comorbid conditions, and severity of illness between the study groups. The manuscript is cogently written and the author’s conclusions are supported by the results. Major limitations of the work, primarily its retrospective design and the insufficient duration of therapy (as compared to the Marik study) in the experimental group, are openly discussed by the authors.
Major Issues
No major issues identified.
Minor Issues
1. Lines 56-59 – please clarify if patients were included based on ICD-10 code for septic shock OR ICD-10 for septic shock + vasopressor use
2. Lines 124-130, Table 1 – please document the duration of vitamin C, thiamine, and hydrocortisone in addition to number of doses in both the text and the table to help clarify duration of treatment for the reader.
3. Lines 128-130, Table 1 – please add the n(%) of patients in each group who received hydrocortisone to table 1
4. Table 1 – consider additional bolded labels within the table for lab values/critical illness scores on study inclusion, treatment timing/duration (i.e., timing of antibiotics, timing of fluids) to ease interpretation of this table for the reader
5. Table 1 – what was the baseline SOFA score of the experimental and control groups?
6. Figure 1 – consider placing this figure earlier in the text , prior to table 2, as it demonstrates the results of the primary outcome analysis of the study and is the main take home point of the manuscript
7. Lines 153-168 – was this subgroup analysis pre-planned? The incorporation of a subgroup analysis into the overall analytic approach should be documented in the methods section.
8. Lines 158-162 – the data regarding TT protocol completion should be described earlier in the results section with the baseline characteristics. The fact that the majority of patients did not complete the full TT protocol is an important limitation of the current work and should be clear to the reader before reviewing figure 1 and tables 2 and 3.
9. Line 160, 177 – is there any data to support the authors claim that therapy was discontinued by the bedside providers due to lack of clinical effect? If not then it would be advisable to avoid commenting on such unobserved behaviors as it could be misinterpreted by a reader to suggest a pre-existing bias of the authors as to the results of this analysis (i.e., the authors felt strongly in advance that this therapy was not beneficial). The data presented here clearly shows no mortality benefit and thus the discussion of the bedside provider’s impression of its efficacy seems unnecessary.
10. Lines 186-190 – the authors identify an important difference between this work and the Marik study, namely that TT therapy may have been used as a “rescue” treatment in the current work as compared to an early treatment in the Marik study. Authors should improve this facet of the discussion by contrasting time to TT exposure in the current work as compared to Marik et al. The authors could consider a second subgroup analysis of patients who initiated TT within 24 hrs (with duration of minimum 24 hrs?) to determine if there is any difference in outcomes within and address this potential caveat directly.
11. Introduction – authors can consider including a reference to a well-written review on the current state of TT evidence authored by Moskowitz et al. (PMID: 30373647) in the introduction.
12. Discussion – fewer patients (40.4% in the control group of this study were exposed to hydrocortisone as compared to the Marik study (~56%). Authors should consider highlighting this in the discussion as the fact that no mortality benefit was seen despite a lesser degree of “steroid crossover” strengthens the conclusions of the current work
13. Authors should include a reference to a recently published retrospective study by Shin et al. (PMID 30654592) that also demonstrated a lack of mortality benefit for TT therapy.
14. Authors may consider acknowledging an active randomized control clinical trial (NCT03380507) being conducted to further address the efficacy of TT therapy.
Author Response

(The authors gave the same response as above.)
